# Effect of Extremely Short-Sized MWCNT as Additive Material in High Surface Area Activated Carbon and Its Enhanced Electrical LIC Performance

**DOI:** 10.3390/molecules27207033

**Published:** 2022-10-18

**Authors:** Md Nazmodduha Rafat, Zambaga Otgonbayar, Sun-Hye Yang, Ick-Jun Kim, Won-Chun Oh

**Affiliations:** 1Department of Advanced Materials Science & Engineering, Hanseo University, Seosan-si 356-706, Chungnam, Korea; 2Korea Electrotechnology Reserch Institute, Next Generation Battery Research Center, 12, Jeongiui-gil, Seongsan-gu, Changwon-si 51543, Gyeongsangnam-do, Korea

**Keywords:** extremely short-sized CNT, high surface area activated carbon, lithium-ion capacitor, electrical performance, specific resistance

## Abstract

Extremely short-sized multi-wall carbon nanotube (CNT) and high surface area activated carbon were used to increase the electrical performance of lithium-ion capacitors (LIC). After electrodes were synthesized using extremely short-sized CNTs and high specific surface area activated carbon, their electrochemical characteristics were evaluated by XRD, SEM, TEM, cyclic voltammetry, EIS, BET, adoption isotherm, t-plot, and pore size distribution. In the process of electrode preparation using extremely short-sized CNTs and high specific surface area activated carbon, CNTs certainly caused a space-filling effect between these two materials, which had a significant effect on the evaluation of electrical characteristics. These relationships were demonstrated by the results of adsorption–desorption isotherm, pore size distribution, t-plot, and BJH plot. Particularly, in the electrochemical cyclic test, as the content of CNT increased, the current density significantly increased with the formation of a near-perfect rectangular shape. This tendency also exhibited excellent characteristics in a t-I plot, Tafel plot, and LSV plot, which clearly affected the electrochemical oxidation–reduction reaction due to the densification of filling density and space structure by adding extremely short-sized CNTs to the active material. In addition, YP80_CNT3 formed a specific resistance value in the range of 7.2 to 6.2 Ω/cm^2^, showing significantly reduced values compared to other samples. This research presented herein offers a promising route for the rational design of MWCNT and stable electrochemical reaction with LIC working mechanism.

## 1. Introduction

Recently, lithium-ion batteries (LIBs) and lithium-ion capacitors (LICs) have required a long life cycle and high power density in terms of portable electronic devices, electronic devices (EVs), hybrid electric devices (HEVs), and large-scale energy storage systems (ESSs) in high-time fields [1,2,3]. Although graphite and graphite composites are widely commercialized as negative electrodes in LIBs and capacitors (LICs), with the advantage of low average voltage, good rate capability, low cost, and tunability of the solid-electrolyte-interphase (SEI) layer, they are not likely to satisfy the increasing demand for next-generation LIBs and capacitors (LICs) that require high power density and long cycle life due to their relatively low practical capacity of about 300 mAh g^−1^. Therefore, it is essential to develop an alternative anode material with a high specific capacity, high energy density, good cyclability, high rate capacity, and high power density. In this regard, activated carbon (YP80) and multi-wall carbon nanotube (MWCNT) have attracted enormous interest as ideal candidates for anode materials due to their high theoretical capacity based on conversion reactions with a high active surface [4,5]. Thus, high surface area activated carbon and carbon nanotube (CNT) have taken prominent main material and additive material replace recently. The good conductivity of electrodes plays an important role in increasing power density.

Activated carbon or modified activated carbon-based lithium-ion capacitors (LICs) have been evaluated for their most important potential for energy storage devices due to their high power density and excellent cycling durability [6,7,8,9,10,11]. The imbalance in kinetic behavior and oxidation–reduction reactions between two electrodes in LICs can result in little improvement in energy and power density simultaneously [12,13,14,15,16,17,18]. However, the intercalation of Li^+^ cations is significantly slower than the adsorption of PF^6^^−^, an anion between the anode and the anode. Therefore, extensive efforts are being made to increase specific capacity and energy density without losing high power density [19,20,21,22,23]. For preferred LICs, the cathode and anode active materials should be synergistically matched. Recently, graphene [24,25], carbon nanosheets [23,26], and carbon nanotubes [27] have been irradiated as cathode and anode active materials to prevent electrode motion imbalance while proving the suitability of a symmetrical battery configuration [28,29]. However, carbon materials still have not shown high energy and output density due to their theoretical capacity upper bound (typically less than 500 mAh g^−1^) and slow reaction kinetics. To improve these characteristics, this research was conducted with activated carbon (YP80) and carbon nitride (MWCNT).

The proper electrochemical properties of LICs depend heavily on the high conductivity and fast Li^+^ diffusion of electrode materials. Therefore, achieving a good rate of performance and desirable transport kinetics for a long cycle life requires a reasonable design and controllable synthesis of carbon electrodes. The electron/ion transfer path of the electrode material mainly includes clearance and vacancy diffusion. Carbon-based materials already have the advantage of epilepsy diffusion due to their inherent layered open framework. As a result, constructing an appropriate empty space in a crystal structure to provide effective spatial diffusion can increase electron/ion transfer in carbon-based electrodes [30,31]. In general, while most prior studies generally focus on carbon restructuring and modification to facilitate reaction dynamics, few studies have studied how to regulate the internal crystal structure of carbon to speed up electron/ion transport. Many reports of amorphous biomass carbon and crystalline carbon have been reported [31,32,33]. Sometimes, the characteristic evaluation results using graphite/polymer composites and graphene/conductive polymer composites can be seen [34,35]. However, few studies have focused on the microcrystalline state. In particular, the use of modified CNTs to increase the conductivity of the electrodes is emerging as a new research field [36,37]. The mechanism that affects the energy-storage ability of microcrystalline carbon in its capacitive coupling state is still unclear [38,39,40,41,42]. YP80 and MWCNT with high active surface areas can be effective in this field. In this regard, we synthesized YP80_CNT1, YP80_CNT2, and YP80_CNT3 (CNT ratio 1:2:3) with a simple mixing method.

A porous activated carbon microcrystalline structure can promote the rapid transfer of electrons/ions with high conductivity. CNTs can stimulate internal field effects to facilitate fast charge transfer and fast diffusion dynamics. The size effect of extremely shot-sized CNTs provides rich reaction sites and induces excessive electrons around local carbon atoms to form a negative charge center, accelerating the transfer of Li^+^ ions. This work will provide future-oriented theoretical support for the optimal design of energy storage materials and the rational generation of high-power capacitors.

## 2. Results and Discussion

### 2.1. Characterization of Active Material

After mixing the high specific surface area activated carbon and the milled CNT at each content for preparing the electrode with lithium metal as a counter electrode, XRD analysis with powder obtained from the electrode was carried out. The results of XRD analysis are presented in Figure 1. This analysis was performed to examine the electrochemical behavior of lithium. As shown in Figure 1, large amounts of lithium compounds such as LiC (JCPDS 14-0649), Li_2_O (JCPDS 09-0355), and Li_2_CO_3_ (JCPDS 22-1141) were formed in activated carbon pores. After electrochemical reactions, samples were dried and analyzed. It was found that large amounts of oxides and carbides were formed on the samples’ surfaces. Thus, it could be concluded that electrode reactions between lithium and activated carbon in the electrolyte were smooth. The diffusion path of lithium ions on the activated carbon surface and the porous core structure can adapt to the intercalation-deintercalation reaction of lithium ions to maintain the stability of the structure, thus improving electrochemical performance [42,43,44].

SEM analysis was performed to study the surface state of the electrode after the electrode reaction; the results are shown in Figure 2. It was found that large amounts of lithium metal and lithium compounds were attached to the electrode. In addition, it was clearly seen that as the content of CNT increased, the amount of exposed activated carbon decreased. As shown in Figure 2a, unreacted activated carbon had a large amount. It was believed that CNTs added for the improvement of conductivity not only reduced spaces between activated carbon particles, but also increased the electrochemical reactivity of lithium between activated carbon particles [44,45,46]. In addition, the decrease in space among activated carbon particles was directly correlated with conductivity. As shown in Figure 2c, many compounds were formed on the electrode surface without many exposed activated carbon particles.

For electrode reaction and nanoscale studies, surface states between activated carbon and CNT were analyzed by TEM and HRTEM. The results are displayed in Figure 3. The location and size of CNTs on the surface of the activated carbon were mainly analyzed. Results showed that the size of CNT was 50 nm to 80 nm. The HRTEM image revealed the network structure of short CNT. As shown in Figure 3a, CNT was spherical and hemispherical with a multilayer network structure. It was confirmed that CNTs added to improve conductivity were closely adhered to the surface of the activated carbon. These results showed that the reactivity between activated carbon and CNT was good. It seemed that the reactivity not only reduced the space between them but also increased electrical conductivity of electrodes themselves. As shown in Figure 3c, the three-dimensional structure of CNT exposed on the activated carbon surface could be clearly seen. In addition, CNT was stably formed on the surface. According to study results, CNTs disperse the nanocarbon sheets and adds to a larger specific surface area, more stable network structure, and more continuous electron transport channels than bare nanocarbon sheet [44,45,46,47]. They have abundant active sites for adsorbing/converting electrons. The internal porous structure and clear outer shell also ensured enough energy storage space with a high lithium ion diffusion effect and a stable structure to achieve outstanding electrochemical performance.

After the electrochemical reaction, surface physicochemical characteristics were analyzed to understand the surface states and pore shape of active materials; the results are shown in Figure 4. Figure 4a shows nitrogen adsorption–desorption isotherms of samples. These adsorption–desorption isotherms showed considerable amounts of adsorption in micropore and macropore regions. Adsorption amount showed little changes in the middle regions of the relative pressure. Hysteresis indicating the presence of mesopores was not significant. These phenomena indicated that micropores formed by activated carbon and macropores formed by small CNT were present on the electrode material’s surface. As shown in Figure 4a, the adsorption amount decreased as the CNT content increased. This indicates that relative adsorption amounts depend on amounts of activated carbon. Figure 4b shows the pore size distribution curve. The size of the pores, pores filled with lithium, and pores not filled with lithium are important for the electrode characteristics. Most of the pores were distributed in the range of 2.38 to 2.87 nm, which mainly consisted of micropores and small amounts of mesopores. Pores of samples had almost similar sizes. As the amount of CNT increased, the pore size gradually increased. Therefore, as predicted from the adsorption isotherm, large pores were formed by CNT. In addition, as the amount of CNT increased, the amount of adsorption decreased along with the large pore size. From the BET adsorption isotherm, we obtained specific surface area values, total pore volume, and mean pore diameter. These results are presented in Table 1.

From these results, it could be seen that the specific surface area value decreased when the content of CNT was increased. In addition, the total pore volume gradually increased. These results showed a consistent tendency with the data values presented above. Mean pore diameter also tended to increase. It was found that CNT had a great influence on pore formation of the electrode active material. It is known that using t-plot is a method of calculating the external surface area, including mesopores and macropores from the slope in a given range using the Harkins–Jura equation. The surface area of the micropore is then obtained by subtracting the external specific surface area from the BET specific surface area. Results are shown in Figure 4c. As shown above, the surface area of micropores decreased as the content of CNT increased. This is consistent with the previous results presented above. Various physical parameter values obtained from the t-plot are shown in Table 2. These results showed that as the content of CNT increased, the external surface area value also increased. On the other hand, total surface area and the micropore surface area values tended to decrease. For the analysis of characteristics of mesopore, the BJH curve is shown in Figure 4d. YP80_CNT3 shown in the adsorption isotherm of Figure 4a implies the distribution of mesopores along with the hysteresis curve. Consistent with these results, increasing the CNT content increased the adsorption amount of mesopores. Therefore, it is believed that CNT will play an important role in the adsorption–desorption of lithium on the activated carbon-CNT electrode surface. Generally, activated carbon has a high surface area with high pore volume amounts. The higher surface area could increase electrochemical properties, and the pore could increase electron excitation, which clarifies sample conductivity [48].

### 2.2. Electrical Performance Test

The results of evaluating electrochemical characteristics are shown in Figure 5. Figure 5a shows the characteristics of a redox reaction cycle, which showed good cycle characteristics for all samples. In particular, among the three samples, Figure 5a showed distinct cycle characteristics along with a large oxidation–reduction curve. It could be seen that CNT had a great influence on current density in the cyclic redox reaction. As shown in the figure, current density values varied greatly depending on the CNT content. Figure 5b shows a graph of the relationship between time and current in the electrochemical redox reaction. Even in these graphs, YP80_CNT3, which had the largest amount of CNT, showed excellent characteristics, consistent with the redox reaction between the current and voltage of Figure 5a. The redox reaction over time also showed many differences in current values depending on the amount of CNT. The relationship with the electrochemical dynamics related to the rate of electrochemical reaction against overvoltage is shown in Figure 5c with a Tafel graph. This figure shows a correlation between the anodic Tafel slope and cathodic Tafel slope. In the case of YP80_CNT1, the cathodic slope showed a relatively stable state, whereas the anodic slope showed an unstable state. It was believed that this represented an unstable chemical reaction of lithium used as an anode. However, it was considered that the cathode using activated carbon by the addition of CNT as an electrode maintained a stable reaction. On the other hand, in the case of YP80_CNT2 and YP80_CNT3, a stable redox reaction was shown between the two electrodes. Therefore, it could be seen that CNT induced a stable electrochemical reaction between the two electrodes with a great influence on the redox reaction. Data from these samples were obtained using linear sweep voltammetry (LSV) to find out the movement (oxidation or reduction) and behavior of electrons accompanied by the CNT-activated carbon electrode material for electrical stimulation. Figure 5d illustrates the results of measuring a current-potential curve by changing the potential of a working electrode from the initial potential E to a constant speed V (V/sec) in a positive or negative direction.

For a reversible electrode reaction, the peak current i_p_ is given as follows:i_p_ = 0.446 nFAC (nF/RT)^1/2^·V^1/2^·D^1/2^(1)
where n is the number of electrons participating in the electrode reaction, F is the Faraday constant (96,487 C/moL), C is the concentration of active species, V is the potential scanning speed, and D is the diffusion coefficient of active species at 25 °C. As shown in the figure, the peak current value increased as the amount of CNT species increased. Therefore, the number of electrons participating in the electrode reaction was increased [45,46,47,48,49,50,51]. As shown in the figure, the peak current value increased with the current density. In the case of YP80_CNT1 and YP80_CNT3, the peak current value was shown directly in proportion to the current density value. The initial peak current value of YP80_CNT2 showed a proportional relationship with the current density value. However, the final normal current value showed an inversely proportional relationship with the current end value. This could be explained by the unstable redox reaction of the electrode itself.

Impedance is expressed as resistance and reactance. These relationships are shown in Figure 6. Impedance values of real and imaginary parts of the three samples are shown in Figure 6a. All three types of samples formed a semi-circle. As the content of CNT increased, the semi-circle became smaller. In addition, as the amount of CNT increased, the impedance value of the imaginary part gradually decreased. Comparing the three types of samples, a constant resistance value can be applied at a constant concentration of electrolyte. Thus, the value presented in the figure can be expressed as a unique impedance value of the electrode sample. These results showed that the addition of CNT had an effect of reducing the electrode resistance and that the resistance value decreased as the content of CNT increased. The intermediate-frequency response is associated with electron transport and transfer at the carbon electrode interface [52]. It was noted that the arc radius of the Nyquist semi-circle of the YP80_CNT3 sample was much smaller than that of YP80_CNT1 or YP80_CNT2. The YP80_CNT3 sample exhibited the smallest arc radius of the Nyquist semi-circle, suggesting that the YP80_CNT3 sample had a lower resistance. These results further confirmed that the incorporation of CNT into the YP80 sample could facilitate electron-hole separation. The variation in specific resistance values over time is shown in Figure 6. It could be seen that as the content of CNT increased, the specific resistance value significantly decreased. It could be seen that YP80_CNT1 had a specific resistance value in the range of 934 to 1496 Ω/cm^2^, which was significantly larger. YP80_CNT2 formed a specific resistance value in the range of 41 to 129 Ω/cm^2^, which was found to be significantly reduced compared to YP80_CNT1. In addition, YP80_CNT3 formed a specific resistance value in the range of 7.2 to 6.2 Ω/cm^2^, which was significantly reduced compared to the values of the previous two samples. These phenomena were clear evidence that the addition of a fairly small size CNT had an effect of densifying the particle structure in the electrode, thus improving the electrical conductivity. The high-specific surface area activated carbon had many pores. Thus, the storage effect of Li ions was large. However, the number of stacked structures was small due to the carbon structure and the fact that the Lc value of the graphene structure was relatively not large. Therefore, these activated carbon properties did not have a great electron transfer effect. As shown in previous SEM images, the activated carbon particle size was 30–50 µm. These particle properties will create many spaces between particles and particles. If these spaces are filled with particulate CNTs with good conductivity, it will have a very good effect in improving conductivity.

Figure 7 shows changes in charge–discharge caused by time and voltage variations. All three types of samples showed the same type of charge-discharge curves. As samples showed a spherical and hemispherical structure, the distance between the YP80 and CNT molecules was low and strongly bonded. Finally, YP80_CNT3 showed a specific capacity effect that was about three times higher than YP80_CNT2 and about five times higher than YP80_CNT1. Their large number of porous behaviors increased the high active surface area and produced higher electron volume essential for LIBs and LICs.

## 3. Experimental

### 3.1. Materials

Activated carbon (YP80) (Kuraray, Tokyo, Japan), MWCNT (US Research Nanomaterials Inc., Houston, TX, USA), polyvinylidene fluoride (PVDF, Sigma-Aldrich Co., St. Louis, MO, USA), and 1-methyl-2-pyrrolidone (99.5%) (Samchun Pure Chemical Co., Ltd., Pyeongtaek, Korea) were purchased. All reagents used in the experiment were of commercial grade.

### 3.2. Synthesis of Extremely Shot-Sized Multi-Wall Carbon Nanotube (CNT) and YP80_CNT

Firstly, extremely shot-sized CNT was made from long hair-like MWCNTs. MWCNT was milled with a planetary Miller for 16 h. After completely making extremely shot-sized CNT powder, it was used to prepare an electrode as an additional material. YP80F was used as a high surface activated carbon source in the synthesis process. YP80 activated carbon, and 1, 2, and 3 wt% activated carbon were mixed to produce active electrode materials, named YP80_CNT1, YP80_CNT2, and YP80_CNT3, respectively. In this preparation, 0.1 g of PVDF as a binder material was dissolved in 10 mL 1-methyl-2-pyrrolidone solution to make a homogenous solution. After that, 0.9 g YP80_CNT was added to that solution and mixed with a homogenous mixer.

### 3.3. Preparation of Electrode

The above-prepared active material was made in a slurry form. This slurry-type active material was coated on a foam-type nickel plate as a thin film type to be used as a cathode. At this time, the active material was uniformly coated to a thickness of 5 µm. The electrode prepared was dried for 8 h at a temperature of 70 °C. A cathode of YP80_CNT and a lithium wire-type anode were installed in the electrolytic container. At this time, 1M LiPF_6_, EC, and EMC were mixed at a ratio of 1:3:6 as the electrolyte. Simple electrode preparation procedure and electrochemical test method are shown in Figure 1.

### 3.4. Characterization

Structural morphologies of crystals of products were analyzed by powder X-ray powder diffraction (XRD; Rigaku, X-beam Diffractometer) with Cu-Kα radiation (λ = 1.5406 A) at 40 kV and 30 mA over a 2-θ range of 20° to 70°. The prepared NLM powder for XRD analysis was acquired using a General Structure Analysis System (GSAS, A.C. Larson and R.B. Von Dreele, Los Alamos National Laboratory). Scanning electron microscopy (SEM JOEL 6701, JEOL Ltd., Tokyo, Japan) images were obtained to clarify the morphology. The size and morphology were examined using a high-resolution transmission electron microscope (HRTEM) and a transmission electron microscope (TEM; JSM-76710F, JEOL, Tokyo, Japan) at a 300 kV excitation voltage. Nitrogen adsorption/desorption isotherms were analyzed by Micro Active for ASAP 2460. Pore-size distribution was figured out from the adsorption–desorption isotherm. At the same time, a t-plot was obtained for evaluating microporous characteristics. A BJH plot was obtained for evaluating mesopore characteristics from the same equipment. Electrochemical characteristics of CV (I-V), EIS, Tafel, and LSV tests were determined or performed with an SP2 electrochemical workstation (Powered by ZIVELAB, Seoul, Korea). Initial potential, final potential, and scan speed are measured and calibrated to electrical properties for measurement methods.

## 4. Conclusions

In summary, this work synthesized high-energy YP80_CNT1, YP80_CNT2, and YP80_CNT3 microcrystalline-activated carbon composites via a simple mixing approach. It also studied the kinetics and lithium storage mechanism of capacitors and batteries to boost lithium-ion diffusion. The effects of both ion diffusion and capacitive behavior mechanisms on lithium storage were investigated by performing kinetic analysis. Porous YP80_CNT1, YP80_CNT2, and YP80_CNT3 layered structure promoted a rapid transfer of electrons/ions and possessed high conductivity, while effects of extremely short CNT particle size on the electrochemical performance of composites were evaluated by first-principle calculations. The porous structure in YP80_CNT1, YP80_CNT2, and YP80_CNT3 stimulated a built-in electric field effect, which facilitated fast charge transport and satisfactory reaction kinetics. The extremely short CNT particle size effect can provide abundant reactive sites and induce excessive electrons around local carbon atoms, forming negative charge centers to accelerate the diffusion of lithium ions. This study offers an in-depth comprehension of electrons/ions diffusion and lithium storage of YP80_CNT1, YP80_CNT2, and YP80_CNT3 electrodes. It provides rational ideas for the optimal construction of high-energy carbon-based materials.

## Data Availability

Data can be made available upon written request to the corresponding author and with a proper justification.

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
