# Peer review of "Effect of Extremely Short-Sized MWCNT as Additive Material in High Surface Area Activated Carbon and Its Enhanced Electrical LIC Performance"

_molecules, 2022, doi:10.3390/molecules27207033_

Round 1

Reviewer 1 Report

Referee report on “Effect of Extremely Short-sized CNT as Additive Material in High Surface Area Activated Carbon and Its Enhanced Electrical LIC Performance” by Md Nazmodduha Rafat et al

This is a quite interesting article that can be recommended for publication, but after clarifying and detailing some parts of the text.

1.     Abstract. Decryption of CNT is required. Not everyone needs to know what this is carbon nanotube.

2.     First paragraph of the introduction. A clearer explanation of novelty and relevance is needed here. All 5 links are over 10 years old and it is not clear what progress has been made in recent years.

3.     Line 70-79. To attract a wider readership, more information about CNT and electrical effects are needed.

Karbovnyk, I., et al. "The impact of temperature on electrical properties of polymer-based nanocomposites." Low Temperature Physics 46.12 (2020): 1231-1234.

Pietrzak, L.; et al. Multiwalled Carbon Nanotubes Polylactide Composites for Electrical Engineering—Fabrication and Electrical Properties. Electronics 202211, 3180.

4.     Lines 91-100. What can be added here about the size and structure of the pores?

5.     Line 139.  “According to study results, there is an overlap of the interfaces between different crystal lattices”. This sentence needs further clarification.

6.     Table 1.  The data on mean pore diameter (nm) need an error bar.

Author Response

Reviewer 1

Referee report on “Effect of Extremely Short-sized CNT as Additive Material in High Surface Area Activated Carbon and Its Enhanced Electrical LIC Performance” by Md Nazmodduha Rafat et al

This is a quite interesting article that can be recommended for publication, but after clarifying and detailing some parts of the text.

  1. Abstract. Decryption of CNT is required. Not everyone needs to know what this is carbon nanotube.

Ans. We revised this part and experimental part as Multi-wall carbon nanotube (CNT).

  1. First paragraph of the introduction. A clearer explanation of novelty and relevance is needed here. All 5 links are over 10 years old and it is not clear what progress has been made in recent years.

Ans. In this part, we revised and suggest our main study points for high conductivity.

  1. Line 70-79. To attract a wider readership, more information about CNT and electrical effects are needed.

Karbovnyk, I., et al. "The impact of temperature on electrical properties of polymer-based nanocomposites." Low Temperature Physics 46.12 (2020): 1231-1234.

Pietrzak, L.; et al. Multiwalled Carbon Nanotubes Polylactide Composites for Electrical Engineering—Fabrication and Electrical Properties. Electronics 202211, 3180.

Ans. We revised this part with recommended references.

  1. Lines 91-100. What can be added here about the size and structure of the pores?

Ans. We revised these parts with some additional sentences.

The size of the pores, pores filled with lithium, and pores not filled with lithium are important for the electrode characteristics. Most of the pores were distributed in the range of 2.38 to 2.87 nm, which mainly consist of micropores and small amounts of mesopores.

  1. Line 139.  “According to study results, there is an overlap of the interfaces between different crystal lattices”. This sentence needs further clarification.

Ans. We clearly revised this part with new references.

  1. Table 1.  The data on mean pore diameter (nm) need an error bar.

Ans. We confirmed error range from original data and revised.

Reviewer 2 Report

This work synthesized high-performance YP80 through simple mixing method like YP80_ CNT1YP80_ CNT2 and YP80_ CNT3 activated carbon composites. The kinetics of lithium ion diffusion promoted by capacitors and batteries and the mechanism of lithium storage were studied. The layered structure of CNT3 promotes the rapid transfer of electrons/ions and has high conductivity. The porous structure in CNT3 stimulates the built-in electric field effect, which promotes the rapid transfer of charge and good reaction kinetics. The extremely short CNT particle size effect can provide rich reaction sites, inducing excess electrons around local carbon atoms, forming negative charge centers, and accelerating the diffusion of lithium ions. YP80 in this study achieves a specific resistance value of 7.2~6.2 Ω/cm2, which was significantly lower than other samples. This work is interesting and it is important for the related field. I think it can be published after some revisions as follows.

1. On page 13, the corner mark for the "6M LiPF6" should be revised.

2. The format of references is inconsistent like references 43, 44 and 45.

3. The article states "Figure 5 (a) shows characteristics of a redox reaction cycle.  Near-perfect rectangular shape was shown in all cycle reactions. As the amount of CNT increased, the rectangular shape was clearly and largely formed in these reactions. It could be seen that CNT had a great influence on current density in the cyclic redox reaction. As shown in the figure, current density values varied greatly depending on the CNT content.” This sentence describes the nearly perfect rectangle of the cyclic voltammetry curve in Figure 5a, but Figure 5a does not show the perfect rectangle. The graphic description does not match.

4. The related references should be mentioned in the introduction like Surf. Interfaces 2022; 30:101874 and Electrochim. Acta 2021; 372:137861.

Author Response

Reviewer 2

his work synthesized high-performance YP80 through simple mixing method like YP80_ CNT1、YP80_ CNT2 and YP80_ CNT3 activated carbon composites. The kinetics of lithium ion diffusion promoted by capacitors and batteries and the mechanism of lithium storage were studied. The layered structure of CNT3 promotes the rapid transfer of electrons/ions and has high conductivity. The porous structure in CNT3 stimulates the built-in electric field effect, which promotes the rapid transfer of charge and good reaction kinetics. The extremely short CNT particle size effect can provide rich reaction sites, inducing excess electrons around local carbon atoms, forming negative charge centers, and accelerating the diffusion of lithium ions. YP80 in this study achieves a specific resistance value of 7.2~6.2 Ω/cm2, which was significantly lower than other samples. This work is interesting and it is important for the related field. I think it can be published after some revisions as follows.

  1. On page 13, the corner mark for the "6M LiPF6" should be revised.

Ans. Thank you for your comment. We revised it. 1M LiPF6, EC, and EMC were ~

  1. The format of references is inconsistent like references 47, 48 and 49.

Ans. We revised this part. [49] is OK. But we revised page part.

[47] Z. Zhang, L. Lu, Z. Lv, Y. Chen, H. Jin, S. Hou, L. Qiu, L. Duan, J. Liu, K. Dai, Porous carbon nitride with defect mediated interfacial oxidation for improving visible light photocatalytic hydrogen evolution, Appl. Catal. B Environ. 232 (2018) 384–390.

[48] Y. Wen, B. Wang, C. Huang, L. Wang, D. Hulicova-Jurcakova, Synthesis of Phosphorus-Doped Graphene and its Wide Potential Window in Aqueous Supercapacitors, Chem. Eur. J. 21 (2015) 80–85.

  1. The article states "Figure 5 (a) shows characteristics of a redox reaction cycle.  Near-perfect rectangular shape was shown in all cycle reactions. As the amount of CNT increased, the rectangular shape was clearly and largely formed in these reactions. It could be seen that CNT had a great influence on current density in the cyclic redox reaction. As shown in the figure, current density values varied greatly depending on the CNT content.” This sentence describes the nearly perfect rectangle of the cyclic voltammetry curve in Figure 5a, but Figure 5a does not show the perfect rectangle. The graphic description does not match.

Ans. Thank you for your comments. We re-write this part.

It showed good cycle characteristics for all samples. In particular, among the three samples, Figure 5 (a) showed distinct cycle characteristics along with a large oxidation-reduction curve.

  1. The related references should be mentioned in the introduction like Surf. Interfaces 2022; 30:101874 and Electrochim. Acta 2021; 372:137861.

Ans. We clearly revised this part with new references.

Sometimes, the characteristic evaluation results using graphite/polymer composites and graphene/conductive polymer composites can be seen. [34-35].

[34] G. Li , M. Ren, H. Zhou, Observably boosted electrochemical performances of roughened graphite sheet/polyaniline electrodes for use in flexible supercapacitors, Surfaces and Interfaces, 30 (2022) 101874-101883.

[35] H. Zhou, M. Ren, and H. Zhai, Enhanced supercapacitive behaviors of poly(3,4-ethylenedioxythiophene)/ graphene oxide hybrids prepared under optimized electropolymerization conditions, Electrochimica Acta, 372, (2021) 137861-137869.

Round 2

Reviewer 1 Report

The authors have successfully improved their original manuscript so now it can be recommended for publication.

Reviewer 2 Report

The authors have well addressed all my comments. Thanks for an interesting paper.

The reference styles of Refs 34-35 should be corrected as follows.

[34] G. Li , M. Ren, H. Zhou, Observably boosted electrochemical performances of roughened graphite sheet/polyaniline electrodes for use in flexible supercapacitors, Surfaces and Interfaces, 30 (2022) 101874.

[35] H. Zhou, M. Ren, and H. Zhai, Enhanced supercapacitive behaviors of poly(3,4-ethylenedioxythiophene)/graphene oxide hybrids prepared under optimized electropolymerization conditions, Electrochimica Acta, 372, (2021) 137861.